# Wild Nutria (*Myocastor coypus*) Is a Potential Reservoir of Carbapenem-Resistant and Zoonotic *Aeromonas* spp. in Korea

**DOI:** 10.3390/microorganisms7080224

**Published:** 2019-07-30

**Authors:** Se Ra Lim, Do-Hun Lee, Seon Young Park, Seungki Lee, Hyo Yeon Kim, Moo-Seung Lee, Jung Ro Lee, Jee Eun Han, Hye Kwon Kim, Ji Hyung Kim

**Affiliations:** 1Infectious Disease Research Center, Korea Research Institute of Bioscience and Biotechnology, Daejeon 34141, Korea; 2Department of Biomolecular Science, KRIBB School of Bioscience, Korea University of Science and Technology (UST), Daejeon 34113, Korea; 3Division of Ecological Conservation Research, National Institute of Ecology, Seocheon 33657, Korea; 4Biological and Genetic Resources Assessment Division, National Institute of Biological Resources, Incheon 22689, Korea; 5Environmental Diseases Research Center, Korea Research Institute of Bioscience and Biotechnology, 125 Gwahak-ro, Daejeon 34141, Korea; 6Laboratory of Aquatic Biomedicine, College of Veterinary Medicine, Kyungpook National University, Daegu 41566, Korea; 7Department of Microbiology, College of Natural Sciences, Chungbuk National University, Cheongju 28644, Korea

**Keywords:** nutria (*Myocastor coypus*), *Aeromonas* spp., antibiotic resistance, carbapenem, *cphA*

## Abstract

The emergence and spread of antibiotic-resistant *Aeromonas* spp. is a serious public and animal health concern. Wild animals serve as reservoirs, vectors, and sentinels of these bacteria and can facilitate their transmission to humans and livestock. The nutria (*Myocastor coypus*), a semi-aquatic rodent, currently is globally considered an invasive alien species that has harmful impacts on natural ecosystems and carries various zoonotic aquatic pathogens. This study aimed to determine the prevalence of antibiotic-resistant zoonotic *Aeromonas* spp. in wild invasive nutrias captured in Korea during governmental eradication program. Three potential zoonotic *Aeromonas* spp. (*A. hydrophila*, *A. caviae*, and *A. dhakensis*) were identified among isolates from nutria. Some strains showed unexpected resistance to fluoroquinolones, third-generation cephalosporins, and carbapenems. In carbapenem-resistant isolates, the *cphA* gene, which is related to intrinsic resistance of *Aeromonas* to carbapenems, was identified, and phylogenetic analysis based on this gene revealed the presence of two major groups represented by *A. hydrophila* (including *A. dhakensis*) and other *Aeromonas* spp. These results indicate that wild nutrias in Korea are a potential reservoir of zoonotic and antibiotic-resistant *Aeromonas* spp. that can cause infection and treatment failure in humans. Thus, measures to prevent contact of wild nutrias with livestock and humans are needed.

## 1. Introduction

The genus *Aeromonas*, which belongs to the family *Aeromonadaceae*, comprises ubiquitous Gram-negative bacilli found in various aquatic environments and organisms [1]. Among 36 recently described species of the genus (http://www.bacterio.net/), several are known as pathogens of cold-blooded animals, including fish and amphibians, and interest in this genus has increased because of its zoonotic potential [2]. In particular, some aeromonads have been recognized as causative agents of human diseases, including gastroenteritis, skin infections, septicemia, peritonitis, pneumonia, and diarrhea [3,4,5]. Of the currently recognized *Aeromonas* species, *A. hydrophila*, *A. caviae*, and *A. veronii* biovar. *sobria* are the most common species known to cause the majority of human infections [6]. Recently, *A. dhakensis* has been considered the principal species causing bacteremia and soft tissue infection [7]. Although the mode of transmission of these pathogens is not clearly understood, recreational or occupational activities in water (e.g., fishing or swimming) and consumption of contaminated food or water are considered potential transmission routes [5]. Recent studies have indicated that domestic and wild animals also can be sources of transmission to humans [8,9,10,11,12,13].

Interest in *Aeromonas* has increased owing to the emergence of strains that are resistant to commercial antibiotics commonly used in aquaculture and veterinary practice [3,14]. The acquisition of antibiotic-resistance genes of diverse environmental origins in this genus poses a serious potential public health risk [15,16]. Although aeromonads resistant to tetracyclines and quinolones have been reported [17,18,19], their intrinsic resistance against *β*-lactam antibiotics is of great concern [16,20]. Most aeromonads produce chromosomally encoded *β*-lactamases, including three principal Ambler classes: class C cephalosporinases, class D penicillinases, and class B metallo-*β*-lactamases (MBLs) [21,22]. Several MBLs, including ImiS [23], ImiH [24], AsbM1 [25], IMP-19 [26], VIM [27], and CphA [28], have been identified in *Aeromonas*, and clinically relevant *Aeromonas* species harboring MBLs are considered a severe public health risk [20]. The best studied MBL gene in the genus *Aeromonas* is *cphA* [28], which encodes a carbapenem-hydrolyzing MBL that has very specific activity towards carbapenems, the last-resort antibiotics selectively applied to treat severe clinical infections [29,30].

The nutria (or coypu, *Myocastor coypus*) is semi-aquatic rodent native to South America. The International Union for Conservation of Nature and the European Union currently consider this animal to be one of the worst invasive alien species globally because it has harmful impacts on native plant biodiversity and natural ecosystems [31,32]. Nutrias were introduced in Korea in 1985 for fur and meat production, but some of the animals escaped into natural habitats and successfully established wild populations [33,34]. Recently, the Korean government has also designated nutria as an alien species and implemented a control and eradication program [35]. The nutria is known as a carrier of various zoonotic aquatic pathogens that can transmit diseases to livestock and humans [36,37,38].

The objective of this study was to evaluate the incidence of zoonotic *Aeromonas* spp. in wild invasive nutrias captured in Korea between 2016 and 2017. Additionally, we aimed to identify potential virulence factors and antimicrobial resistance mechanisms in *Aeromonas* isolates. Our study findings emphasize the need for predicting and preventing the spread of antibiotic-resistant pathogenic aeromonads and to implement the “One Health” approach to emerging public health threats. To the best of our knowledge, this is the first report to assess the potential virulence and antibiotic resistance in *Aeromonas* spp. isolated from wild nutrias.

## 2. Materials and Methods

### 2.1. Bacterial Isolation and Culture Conditions

Between 2016 and 2017, fresh carcasses of 26 wild nutria (*Myocastor coypus*), which were captured throughout the tributary of Nakdong River (35°19′16.7″N 128°48′25.0″E), were supplied by a network of hunters in Gimhae (Gyeongnam province, South Korea) in the context of an eradication program of the Korean government. Sterile swabs were used to collect specimens from the external wounds, nasal, and rectal cavities of the animals. Bacteria were isolated using a standard dilution plating technique on 5% sheep blood agar (BA; Synergy Innovation, Seongnam, Korea) by incubating them at 37 °C for 24 h. To assess strain purity, single colonies were selected and subcultured three times, and then, the isolated bacteria were identified by 16S rDNA sequencing (Macrogen Inc., Seoul, Korea). Biochemical characteristics of isolates identified as members of the genus *Aeromonas* (Table 1) were analyzed using the API 20E system (bioMérieux, Marcy l’Étoile, France) following the manufacturer’s protocol. All confirmed *Aeromonas* isolates were stored in tryptic soy broth (Difco, Detroit, MI) with 10% glycerol at −80 °C until use.

### 2.2. Species Discrimination

*Aeromonas* isolates were cultured overnight on BA at 37 °C. Bacterial genomic DNA was isolated using a DNeasy Blood & Tissue kit (Qiagen Korea Ltd., Seoul, Korea) following the manufacturer’s protocol. For species discrimination, first, the *gyrB* gene, which encodes DNA gyrase subunit B, was amplified and sequenced using the primers gyrB3F/gyrB14R [39]. Second, the *rpoB* gene, which encodes the β-subunit of DNA-dependent RNA polymerase, was amplified and sequenced using the primers Pasrpob-L/Rpob-R [40]. The sets of primers used for amplification and sequencing of *gyrB* and *rpoB* are listed in Appendix A. The *gyrB* and *rpoB* sequences of the isolates were, respectively, compared with representative sequences from each type strain of *Aeromonas* species in the GenBank database by BLAST searches (www.ncbi.nlm.nih.gov/BLAST). In addition, the *gyrB* sequences of the isolates were aligned with representative sequences from each type strain of *Aeromonas* species using ClustalX (version 2.1) [41] and BioEdit Sequence Alignment Editor (version 7.1.0.3) [42]. Then, the datasets were phylogenetically analyzed using the MEGA ver. 7.0 [43]. A neighbor-joining phylogenetic tree was constructed using a Jukes–Cantor distances matrix, and the reliability of the tree was assessed using 1,000 bootstrap replicates. Finally, 14 *Aeromonas* isolates were identified to the species level.

### 2.3. Determination of Virulence-Associated Genes

To evaluate the pathogenic potential of the *Aeromonas* isolates, several PCR-based methods were used to determine the distribution of the genes coding for cytotoxic heat-labile enterotoxin (*act*, also known as aerolysin/hemolysin), serine protease (*aspA*), heat-labile (*alt*) and heat-stable (*ast*) cytotoxins, components of the type 3 (*aexT* and *ascV*) and type 6 (*vasH*) secretion systems, lateral (*lafA*) and polar (*flaA*) flagella, bundle-forming pilus (*BfpA* and *BfpG*) and Shiga-like toxin (*stx-1* and *stx-2*), as previously described [15]. The sets of primers used for amplification and sequencing of these genes are listed in Appendix A. PCR conditions for gene amplification were based on previous studied referenced in Appendix A. Strains yielding amplicons of the expected size were sequenced and the sequences were compared to the GenBank database.

### 2.4. Antimicrobial Susceptibility Testing

Antimicrobial susceptibility of the isolated aeromonads was evaluated by the disk diffusion method according to the guidelines of the Clinical and Laboratory Standards Institute (CLSI) [44,45]. In total, 20 antimicrobial agents (Oxoid Ltd., Basingstoke, UK) of nine classes were used: penicillins and β-lactam/β-lactamase inhibitor combinations (ampicillin/sulbactam (10/10 μg), amoxycillin/clavulanic acid (20/10 μg), and piperacillin/tazobactam (100/10 μg)), cephems (cephalothin (30 μg), cephazolin (30 μg), cefoxitin (30 μg), cefuroxime (30 μg), ceftazidime (30 μg), cefotaxime (30 μg), and cefepime (30 μg)), carbapenems (imipenem (10 μg) and meropenem (10 μg)), a monobactam (aztreonam (30 μg)), aminoglycosides (amikacin (30 μg) and gentamicin (10 μg)), a tetracycline (tetracycline (30 μg)), fluoroquinolones (ciprofloxacin (5 μg) and levofloxacin (5 μg)), folate pathway inhibitors (trimethoprim/sulphamethoxazole (1.25/23.75 μg)), and a phenicol (chloramphenicol (30 μg)). The minimum inhibitory concentrations (MICs) of six selected antimicrobial agents (amoxicillin (256–0.015 μg), amoxycillin/clavulanic acid (256–0.015 μg), ampicillin (256–0.015 μg), cefotaxime (256–0.015 μg), imipenem (32–0.002 μg), and meropenem (32–0.002 μg)) were determined using MIC Evaluator Strips (Oxoid Ltd.). Inhibition zones and MICs were interpreted based on CLSI guidelines [44,45]. For quality control, *Escherichia coli* ATCC 25922 and ATCC 35218 were used.

### 2.5. Determination of Antibiotic Resistance Genes and Phylogenetic Analysis of the cphA Gene

The genetic determinants associated with resistance to tetracycline, quinolones, *β*-lactams, cephalosporins, and carbapenems in the 14 *Aeromonas* isolates were investigated by PCR analyses. In addition, the isolates were screened for the existence of class 1 integrons, which are gene cassettes encoding resistance to various antimicrobials. The sets of primers used for amplification and sequencing of these genes are listed in Appendix A. Nucleotide and deduced amino acid sequences of the *cphA* genes of the *Aeromonas* isolates were compared and aligned with representative *cphA* variants in Aeromonadaceae available from GenBank, including *cphA* (GenBank accession no. X57102), *cphA2* (U60294), *cphA3* (AY112998), *cphA4* (KM609958), *cphA5* (KP771880), *cphA6* (AY227052), *cphA7* (AY227053), and *cphA8* (AY261375). The deduced amino acid sequences of the *cphA* genes were phylogenetically analyzed using the maximum-likelihood (ML) method with 1,000 bootstrap replicates in MEGA ver. 7.0 [43]. An ML tree was constructed using the suggested WAG+G model with the option of complete deletion of gaps and missing data.

### 2.6. Nucleotide Sequence and Strain Deposition

All *gyrB* and *cphA* nucleotide sequences of the *Aeromonas* isolates in this study have been deposited in GenBank database under accession numbers MK495855–MK495868 and MK415746–MK415756, respectively. A living axenic culture of each of the 14 *Aeromonas* isolates has been deposited in the Korean Culture Center of Microorganisms (KCCM); the accession numbers are provided in Table 1.

## 3. Results and Discussion

Several *Aeromonas* spp. have been reported as important zoonotic pathogens based on their virulence and antibiotic resistance profiles [2]. Although the mode of transmission is not completely understood, recent studies have indicated that domestic and wild animals are potential sources of transmission to humans [8,9,10,11,12,13]. Therefore, the present study aimed to evaluate the incidence of potential zoonotic *Aeromonas* spp. in wild invasive nutria thriving in Korea and to investigate their virulence and antibiotic resistance to allow the development of measures to effectively control dissemination to humans.

### 3.1. Wild Nutria Is a Potential Reservoir of Zoonotic Aeromonas spp.

Twenty-six fresh carcasses of wild nutria captured in a tributary of Nakdong River (Gyeongnam province, Korea) in the context of an eradication program of the Korean government between 2016 and 2017 were supplied by a network of hunters. Bacterial strains isolated from external wounds, nasal, and rectal cavities of the animals were analyzed by biochemical tests (Appendix A) and 16S rDNA sequencing. In total, 14 β-hemolytic isolates were identified as members of the genus *Aeromonas*. Although several studies on *Aeromonas* in wild and domestic animals have revealed its ecological importance as a potential zoonotic pathogen, the identification of the genus in those studies was mainly based on phenotypic traits and 16S rDNAs of the isolates [8,9,46,47] and did not fully reflect its recent taxonomy based on housekeeping genes [48]. Therefore, we further sequenced the housekeeping genes (*gyrB* and *rpoB*) of *Aeromonas* isolates to clarify its taxonomical positions. Based on species discrimination of the aeromonads on the basis of *gyrB* and *rpoB* sequence comparisons, four *Aeromonas* species (*A. hydrophila* (*n* = 10), *A. caviae* (*n* = 2), *A. dhakensis* (*n* = 1), and *A. rivipollensis* (*n* = 1)) were identified (Table 1 and Figure 1). Several wild animals, including fish, amphibians, waterfowls, terrestrial mammals, and various aquatic invertebrates have been reported as potential reservoirs of zoonotic *Aeromonas* spp. [8,9,10,11,12,13]. However, research on their occurrence in wild aquatic or semi-aquatic mammals is relatively scarce.

Although little is known about the microbial flora of nutria, these species have aquatic habitats in both freshwater and marine environments and in fact, several bacterial species normally found in fresh and seawater including *Aeromonas* spp. (e.g., *A. veronii*) have been isolated from frozen and fresh nutria carcasses processed in the USA [46]. Moreover, *A. hydrophila* has been identified as the cause of death in reared nutrias in German nutria farms [49]. Additionally, several *Aeromonas* spp. have been reported as bacterial flora that can cause opportunistic infections in other semi-aquatic mammals, such as the Canadian beaver (*Castor canadensis*) [50] and Eurasian otter (*Lutra lutra*) [47]. In this study, we isolated several *Aeromonas* species that are known to cause human infections from wild nutria in Korea, and especially, *A. hydrophila* was isolated from all swab sample types (rectal and nasal cavities and external wounds). Based on these results, *Aeromonas* spp. could be considered as bacterial flora of wild nutria, similar to other semi-aquatic mammals, and are a potential reservoir of pathogenic *Aeromonas* spp. that can cause opportunistic zoonotic infections in livestock and humans.

Next, we investigated the presence of 13 virulence-related genes in the 14 *Aeromonas* isolates (Table 2 and Appendix A). Overall, *A. hydrophila* and *A. dhakensis* were found to be more virulent than the two other *Aeromonas* spp. in our study. The most prevalent virulence genes were *ast* (14/14), *flaA* (14/14), and *alt* (13/14), whereas *aexT* (3/14) was the least prevalent. Genes encoding the type 3 secretion system (*ascV*), bundle-forming pilus (*BfpA* and *BfpG*), and Shiga-like toxin (*stx-1* and *stx-2*) were not detected in any strain. Exotoxins, including cytotoxic heat-labile enterotoxin (act) [51], cytotonic heat-labile enterotoxin (alt) [52], and cytotonic heat-stable enterotoxin (ast) [53], are major virulence factors of *Aeromonas* spp. In this study, ten strains (9 *A. hydrophila* and 1 *A. dhakensis*) encoded all three enterotoxins, and *A. caviae* and the other one *A. hydrophila* strains possessed *alt* and *ast* genes. The detection of multiple enterotoxin genes, which are significantly associated with gastroenteritis and diarrheal disease [54], may imply that these *Aeromonas* strains can affect humans as an enteropathogen. Moreover, the high prevalence of *flaA*, which is involved in lateral flagella production and biofilm formation in *Aeromonas* [55], in our isolates strongly suggests that they would be able form biofilms in animals and humans. Biofilm formation is associated with difficult-to-treat persistent infection and chronic inflammation [56]. The strain KN-Mc-11N1, which was previously reported as *A. rivipollensis* [57], possessed the least number of virulence genes among the *Aeromonas* strains examined in this study.

### 3.2. Wild Nutria Is a Potential Reservoir of Antimicrobial-Resistant Aeromonas spp.

The significance of antibiotic-resistant *Aeromonas* spp. in disease outbreaks in aquaculture has been relatively well investigated because of the emergence of resistance to commercial antibiotics, such as tetracyclines and quinolones [17,18,19]. However, the intrinsic resistance of non-aquaculture-originated pathogenic aeromonads against *β*-lactam antibiotics such as cephalosporines and carbapenems remains a great concern because of the potential public health risk [20,21]. Therexfore, we investigated the resistance phenotypes of the 14 *Aeromonas* isolates to several antibiotic classes and were attempted to uncover the genetic determinants. Antimicrobial susceptibility was evaluated by the disc diffusion method (Table 3) and MIC determination (Table 4). According to the results of the disc diffusion assay, most isolates were not resistant to monobactams, aminoglycosides, tetracyclines, fluoroquinolones, folate pathway inhibitors, and phenicols. We did detect isolates resistant to *β*-lactam/*β*-lactamase inhibitor combinations, and first- and second-generation cephalosporins and carbapenems. Only *A. hydrophila* strains KN-Mc-5R1 and KN-Mc-6U22 showed resistance to monobactam (aztreonam) and fluoroquinolone (levofloxacin), respectively. Interestingly, *A. hydrophila* strains KN-Mc-4N1 and KN-Mc-5R1 were resistant to ceftazidime, a third-generation cephalosporin (Table 3). Third-generation cephalosporins have demonstrated excellent antimicrobial activity against *Aeromonas* species in clinical infections [58,59,60,61]; however, the emergence of resistance against those antibiotics in human isolates has been recently reported [62].

Unexpectedly, regardless of the low rates, the emergence of third-generation cephalosporin-resistant zoonotic bacteria has been recorded in terrestrial wild animals from Italy [63] and the USA [64]. *Aeromonas* is not the exception, and actually, third-generation cephalosporin-resistance was also recorded in some isolates from Eurasian otters in Portugal [65]. Similarly, two *Aeromonas* isolates in this study showed resistance to third-generation cephalosporin. These findings clearly indicate that environmental- or wild animal-originating *Aeromonas* spp. have already acquired resistance to these antibiotics and might pose a serious public health risk in different regions including Korea. Moreover, all *A. hydrophila* strains, except strain KN-Mc-6U2, were completely or intermediately resistant to imipenem, one of the carbapenems that are selectively used as last-resort antibiotics in humans, and particularly, *A. dhakensis* strain KN-Mc-6U21 was strongly resistant to both imipenem and meropenem (Table 3 and Table 4). Overall, *A. hydrophila* and *A. dhakensis* were more resistant to *β*-lactam antibiotics than the other *Aeromonas* species investigated in this study, and most strains were multi-drug resistant.

The occurrence of antibiotic-resistance genes associated with tetracycline, quinolone, *β*-lactam, cephalosporin, and carbapenem resistance, as well as class 1 integrons, in all 14 *Aeromonas* isolates was investigated by PCR and no positive amplicons were detected in this study. Although some *A. hydrophila* strains (KN-Mc-4N1 and KN-Mc-5R1) were phenotypically resistant to third-generation cephalosporin, we were not able to detect several chromosome- and plasmid-mediated extended-spectrum β-lactamases (ESBL)-related genetic determinants, which are mainly disseminated in livestock and humans [66]. Several studies have reported that *Aeromonas* spp. are uniformly and intrinsically resistant to *β*-lactam antibiotics due to the production of multiple inducible, chromosomally-encoded *β*-lactamases [67,68]. Based on these results, we assume that the third-generation cephalosporin resistance of our isolates might be associated with another unveiled chromosomal AmpC *β*-lactamase in *Aeromonas*, which can hydrolyze third-generation cephalosporins, and further studies are warranted in our future analysis.

The *cphA* gene, related to carbapenem resistance [28], was detected in 11 out of 14 isolates (78.5%), but it was not detected in *A. caviae* and *A. rivipollensis*. Carbapenem resistance due to CphA, encoded by *cphA*, is known to be prevalent, but species-related in *Aeromonas* [69]. To date, the *cphA* gene has been found in *A. dhakensis*, *A. hydrophila*, *A. veronii*, *A. jandaei*, and *A. salmonicida*, but not in *A. caviae* [70,71]. In accordance with these previous reports, none of our *A. caviae* isolates (KN-Mc-1R3 and KN-Mc-3R1) showed phenotypical resistance to carbapenems or possessed the *cphA* gene. One of the *cphA*-encoding *A. hydrophila* isolates, KN-Mc-6U2, also did not show phenotypical resistance to carbapenems. The *cphA* genes of the 11 isolates showed 96–98% identity to the reported *A. hydrophila cphA* gene (X57102). Deduced amino acid sequences were phylogenetically analyzed in comparison with eight representative *cphA* variants found in Aeromonadaceae available from GenBank. The phylogenetic tree revealed two major groups (Figure 2). The *cphA* genes of all *A. hydrophila* isolates were clustered together with the *cphA* genes from other *A. hydrophila* strains (*cphA,* X57102; *cphA2*, U60294; *cphA5*, KP771880). The nucleotide sequence of *cphA* of *A. dhakensis* KN-Mc-6U21 was very similar (97.0%) to that of the type strain *A. dhakensis* MDC67^T^ (AB765398), and interestingly, was clustered with the *A. hydrophila cphA* genes, different from those of the other *Aeromonas* species. Unlike the carbapenem-resistant *A. hydrophila* isolates in this study and despite the high *cphA* sequence similarity, *A. dhakensis* KN-Mc-6U21 showed extended resistance to imipenem and meropenem. Based on these results, it can be assumed that the *cphA* genes of *A. hydrophila* and *A. dhakensis* are distinct from those of other *Aeromonas* species; however, given the limited number of *A. dhakensis* isolates in this study, further studies on the genetic characteristics and diversity related to phenotypical carbapenem resistance are needed.

*A. dhakensis* reportedly has caused fatal animal and human infections in Korea [72,73,74]; however, the emergence of carbapenem resistance had not been reported to date. In this study, we detected a virulent *A. dhakensis* isolate (*act*+/*alt*+/*ast*+) that was simultaneously resistant to imipenem (32 μg/mL) and meropenem (8 μg/ml). This finding suggest that wild nutrias in Korea can carry potential pathogenic *Aeromonas* spp. including *A. dhakensis* that can cause potential zoonotic infections and might lead to treatment failure when using carbapenems in humans. Nutrias are known as carriers of various zoonotic aquatic pathogens that can cause diseases in livestock and humans, and *Aeromonas* spp. can be considered one of these potential zoonotic pathogens based on our findings. Although wild nutria are regarded an alien invasive species in Korea and the Korean government has implemented a control and eradication program [75], additional measures to prevent contact with livestock and humans will have to be developed as wild nutria meat and byproducts are currently still improperly consumed in Korea.

In conclusion, the results of this study support earlier findings that wild nutria can serve as a reservoir of various zoonotic aquatic pathogens that can transmit diseases to livestock and humans. Several potential zoonotic *Aeromonas* strains that showed unexpected resistance to antibiotics used in human and veterinary medicine were isolated from wild nutria captured in Korea. The intrinsic carbapenem resistance gene, *cphA*, was identified in most isolates, and phylogenetic analysis revealed the presence of two major groups of the genetic determinants. These results indicate that wild nutrias in Korea are a potential reservoir of zoonotic and antibiotic-resistant *Aeromonas* spp. that can cause infection and treatment failure in humans. Thus, additional measures to prevent contact of these wild animals with livestock and humans will have to be developed.

## Figures and Tables

**Figure 1 microorganisms-07-00224-f001:**
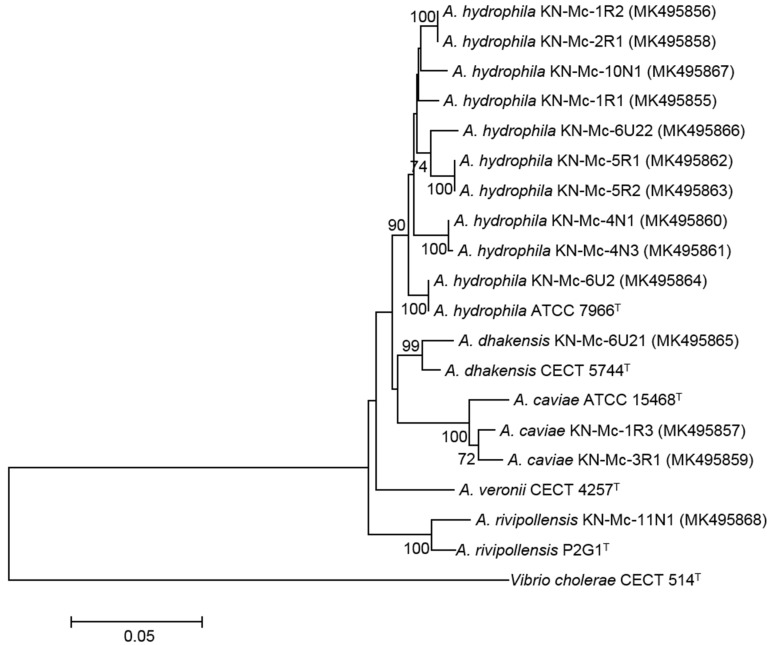
Neighbor-joining phylogenetic tree based on *gyrB* nucleotide sequences showing the relationships of all *Aeromonas* isolates reported in this study to some representative type strains of *Aeromonas* spp. and the outgroup *Vibrio cholerae* CECT 514^T^. The scale bar represents 0.05 nucleotide substitutions per site.

**Figure 2 microorganisms-07-00224-f002:**
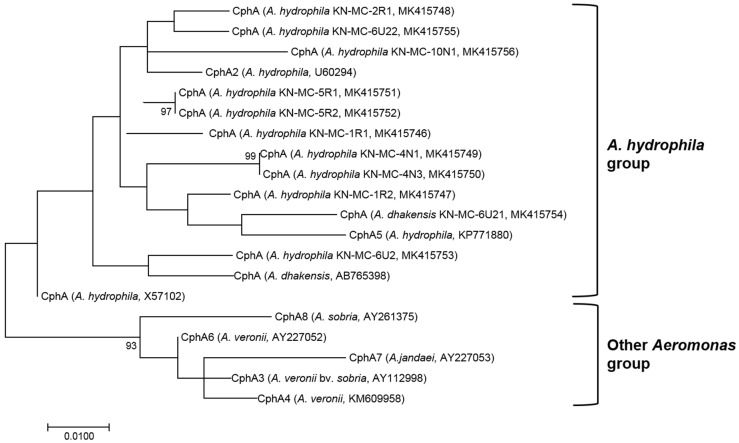
Maximum-likelihood phylogenetic tree based on *cphA* amino acid sequences showing the relationships of the deduced amino acid sequences of 11 *cphA* genes reported in this study to eight representative *cphA* variants in Aeromonadaceae. The scale bar represents 0.01 amino acid substitutions per site.

**Table 1 microorganisms-07-00224-t001:** *Aeromonas* isolates identified in this study.

No.	Bacterial Strains	Hemolysis	Isolated Year	Source	Deposition Number **
1	*Aeromonas hydrophila* KN-Mc-1R1	β	2016	Rectal cavity	KCCM 90327
2	*Aeromonas hydrophila* KN-Mc-1R2 *	β	2016	Rectal cavity	KCCM 90286
3	*Aeromonas cavieae* KN-Mc-1R3	β	2016	Rectal cavity	KCCM 90328
4	*Aeromonas hydrophila* KN-Mc-2R1	β	2016	Rectal cavity	KCCM 90329
5	*Aeromonas cavieae* KN-Mc-3R1	β	2016	Rectal cavity	KCCM 90330
6	*Aeromonas hydrophila* KN-Mc-4N1	β	2016	Nasal cavity	KCCM 90331
7	*Aeromonas hydrophila* KN-Mc-4N3	β	2016	Nasal cavity	KCCM 90332
8	*Aeromonas hydrophila* KN-Mc-5R1	β	2016	Rectal cavity	KCCM 90333
9	*Aeromonas hydrophila* KN-Mc-5R2	β	2016	Rectal cavity	KCCM 90334
10	*Aeromonas hydrophila* KN-Mc-6U2	β	2016	External wound	KCCM 90335
11	*Aeromonas dhakensis* KN-Mc-6U21 *	β	2016	External wound	KCCM 90283
12	*Aeromonas hydrophila* KN-Mc-6U22	β	2016	External wound	KCCM 90336
13	*Aeromonas hydrophila* KN-Mc-10N1	β	2017	Nasal cavity	KCCM 90337
14	*Aeromonas rivipollensis* KN-Mc-11N1 *	β	2017	Nasal cavity	KCCM 90285
15	*Aeromonas hydrophila* ATCC 7966	–	–	–	

* The genomes of strains KN-Mc-1R2, KN-Mc-6U21, and KN-Mc-11N1 have been deposited in GenBank under accession nos. CP027804.1, CP023141.1, and CP027856.1, respectively. ** KCCM, Korean Culture Center of Microorganisms.

**Table 2 microorganisms-07-00224-t002:** Presence of the virulence-related genes of 14 *Aeromonas* strains.

Strains	Virulence-Related Genes												
	*act*	*aspA*	*alt*	*ast*	*aexT*	*ascV*	*vasH*	*lafA*	*flaA*	*BfpA*	*BfpG*	*stx-1*	*stx-2*
KN-Mc-1R1	+	+	+	+	−	−	+	+	+	−	−	−	−
KN-Mc-1R2	+	+	+	+	−	−	+	+	+	−	−	−	−
KN-Mc-1R3	−	−	+	+	−	−	+	+	+	−	−	−	−
KN-Mc-2R1	−	+	+	+	+	−	+	+	+	−	−	−	−
KN-Mc-3R1	−	−	+	+	−	−	−	+	+	−	−	−	−
KN-Mc-4N1	+	+	+	+	−	−	+	+	+	−	−	−	−
KN-Mc-4N3	+	+	+	+	−	−	+	+	+	−	−	−	−
KN-Mc-5R1	+	+	+	+	−	−	+	+	+	−	−	−	−
KN-Mc-5R2	+	+	+	+	−	−	+	+	+	−	−	−	−
KN-Mc-6U2	+	+	+	+	−	−	+	−	+	−	−	−	−
KN-Mc-6U21	+	−	+	+	+	−	+	−	+	−	−	−	−
KN-Mc-6U22	+	+	+	+	−	−	+	−	+	−	−	−	−
KN-Mc-10N1	+	+	+	+	−	−	+	+	+	−	−	−	−
KN-Mc-11N1	−	−	−	+	+	−	−	−	+	−	−	−	−

**Table 3 microorganisms-07-00224-t003:** Antibiotic resistance profile of the 14 *Aeromonas* isolates as determined by disk diffusion testing.

Strains	Antimicrobial agent [disk content (μg)]																			
	*β*-lactams			Ceph							Carb		Mo	Am		Tet	Fq		F	P
	SAM (20)	AMC (30)	TZP (110)	KF (30)	KZ (30)	FOX (30)	CXM (30)	CAZ (30)	CTX (30)	FEP (30)	IPM (10)	MEM (10)	ATM (30)	AK (30)	CN (10)	TE (30)	CIP (5)	LEV (5)	STX (25)	C (30)
KN-Mc-1R1																				
KN-Mc-1R2																				
KN-Mc-1R3																				
KN-Mc-2R1																				
KN-Mc-3R1																				
KN-Mc-4N1																				
KN-Mc-4N3																				
KN-Mc-5R1																				
KN-Mc-5R2																				
KN-Mc-6U2																				
KN-Mc-6U21																				
KN-Mc-6U22																				
KN-Mc-10N1																				
KN-Mc-11N1																				

A category of antibiotic susceptibility is dark gray, resistant; light gray, intermediate; white, susceptible. β-lactams, β-lactam/β-lactamase inhibitor combinations; Ceph, Cephalosporins; Carb, Carbapenems; Mo, Monobactams; Am, Aminoglycosides; Tet, Tetracyclines; Fq, Fluoroquinolones; F, Folate pathway inhibitors; P, Phenicols. SAM, Ampicillin-Sulbactam; AMC, Amoxycillin-Clavulanic acid; TZP, Piperacillin-Tazobactam; KF, Cephalothin; KZ, Cephazolin; FOX, Cefoxitin; CXM, Cefuroxime; CAZ, Ceftazidime; CTX, Cefotaxime; FEP, Cefepime; IPM, Imipenem; MEM, Meropenem; ATM, Aztreonam; AK, Amikacin; CN, Gentamicin; TE, Tetracycline; CIP, Ciprofloxacin; LEV, Levofloxacin; STX, Trimethoprim-sulphamethoxazole; C, Chloramphenicol.

**Table 4 microorganisms-07-00224-t004:** MICs of six selected antimicrobial agents for the 14 *Aeromonas* isolates.

Strains	Antimicrobial Agent (MIC (μg/mL))					
	*β*-Lactam/*β*-Lactamase Inhibitor Combinations			Cephalo-Sporins	Carbapenems	
	AM	AMC	AMP	CTX	IPM	MEM
KN-Mc-1R1	128 (R)	16 (I)	>256 (R)	0.12	2 (I)	0.06
KN-Mc-1R2	256 (R)	64 (R)	>256 (R)	0.06	8 (R)	1
KN-Mc-1R3	128 (R)	16 (I)	>256 (R)	0.25	0.5	0.015
KN-Mc-2R1	>256 (R)	32 (R)	>256 (R)	0.06	8 (R)	0.5
KN-Mc-3R1	128 (R)	16 (I)	>256 (R)	0.12	0.5	0.03
KN-Mc-4N1	128 (R)	16 (I)	>256 (R)	0.12	2 (I)	0.12
KN-Mc-4N3	64 (R)	32 (R)	>256 (R)	0.06	2 (I)	0.12
KN-Mc-5R1	256 (R)	32 (R)	>256 (R)	0.12	8 (R)	0.25
KN-Mc-5R2	256 (R)	32 (R)	>256 (R)	0.06	4 (R)	0.03
KN-Mc-6U2	64 (R)	16 (I)	>256 (R)	0.06	0.5	0.03
KN-Mc-6U21	128 (R)	16 (I)	>256 (R)	0.5	32 (R)	8 (R)
KN-Mc-6U22	32 (R)	16 (I)	>256 (R)	0.06	2 (I)	0.06
KN-Mc-10N1	64 (R)	16 (I)	>256 (R)	0.06	4 (R)	0.06
KN-Mc-11N1	128 (R)	32 (R)	>256 (R)	0.25	0.12	0.03

R, resistant; I, intermediate. AM, Amoxycillin; AMC, Amoxycillin-Clavulanic acid; AMP, Ampicillin; CTX, Cefotaxime; IPM, Imipenem; MEM, Meropenem.

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
