# Peer review of "Wild Nutria (Myocastor coypus) Is a Potential Reservoir of Carbapenem-Resistant and Zoonotic Aeromonas spp. in Korea"

_microorganisms, 2019, doi:10.3390/microorganisms7080224_

Round 1

Reviewer 1 Report

The manuscript ID: microorganisms-552864 entitled Title: Wild nutria (Myocastor coypus) is a potential reservoir of carbapenem-resistant and zoonotic Aeromonas spp. in Korea deals with studies, which aimed to determine the prevalence of antibiotic-resistant (potentially zoonotic) Aeromonas spp. in wild nutrias captured in Korea during governmental eradication program. These results indicate that wild nutrias in Korea are a potential reservoir of antibiotic-resistant Aeromonas spp.

The paper contains interesting studies, which indicate that wild nutrias in Korea are a potential reservoir of zoonotic Aeromonas spp. Moreover, what is worth highlighting, not only phenotypic but also determination of virulence-associated genes (by PCR) as well as phylogenetic analysis of the cphA gene have been performed. Some nucleotide sequences (gyrB and cphA) have been deposited in the GenBank database.

The writing is good and the data are presented properly in a clear and concise form.

The manuscript can be accepted for publication in Microorganism Journal after the following issues have been addressed. The manuscript needs Minor revision.

Comments:

1)      Materials and Methods, lines 99 and 110-111; and Results and Discussion; lines 183-184,

The isolated bacteria were identified by 16S rRNA sequencing ….

The authors used DNeasy Blood and Tissue kit to isolate bacterial genomic DNA and strains were identified by 16S rDNA/not rRNA sequencing.

2)      Materials and Methods, lines 97-104; Please rearrange/reorder this fragment of the text. First isolated colonies should be selected, purified from single colonies subcloned, and then identified by 16S rDNA sequencing.

3)      Results and Discussion, Supplementary Table S1; The sizes of amplified PCR products(gene fragments) should be inserted in the Table S1. 

4)      Results and Discussion, lines 206-221; The electrophoregrams showing the presence of amplified fragments of virulence-related genes should be inserted in the Supplementary.

5)      Results and Discussion, lines 175-178; and lines 300-301; Therefore, the present study aimed to evaluate the incidence of zoonotic Aeromonas spp. In the Reviewer’s opinion this statement is exaggerated and should be modified. The incidence of potentially zoonotic Aeromonas sp.

6)      Results and Discussion, and lines 300-301; similarly as above.

7)      Table 4. R, I – under the Table should be explained.

8)      Results and Discussion, lines 267-268; Based on the absence of plasmid-mediated genetic determinants….Does this statement find support in the presented results? If does not please add appropriate literature positions.

9)      Results and Discussion. This paragraph should be modified. In the Reviewer’s opinion the results should be separated from the part constituting the discussion part.

Editorial mistakes:

Title; and line 154, : Myocastor coypus, Aeromonas; should be written in italic font.  

Introduction; Please cut the instruction for introduction part of the manuscript.

Author Response

<Response to reviewers: Ms. Ref. No. microorganisms-552864>

On behalf of all authors, I greatly appreciate the helpful advice and comments from ‘Microorganisms’ regarding our manuscript (microorganisms-552864) entitled "Wild Nutria (Myocastor coypus) is a Potential Reservoir of Carbapenem-resistant and Zoonotic Aeromonas spp. in Korea". According to the advice and comments of the reviewers, we have revised the manuscript, and all of our corrections are highlighted with yellow in the manuscript. We hope that the manuscript is now suitable for publication.

Ji Hyung Kim

--------------------------------------------------------------------------------------------------------------

REVIEWERS' COMMENTS TO AUTHORS

<Reviewer #1>

The manuscript ID: microorganisms-552864 entitled Title: Wild nutria (Myocastor coypus) is a potential reservoir of carbapenem-resistant and zoonotic Aeromonas spp. in Korea deals with studies, which aimed to determine the prevalence of antibiotic-resistant (potentially zoonotic) Aeromonas spp. in wild nutrias captured in Korea during governmental eradication program. These results indicate that wild nutrias in Korea are a potential reservoir of antibiotic-resistant Aeromonas spp.

The paper contains interesting studies, which indicate that wild nutrias in Korea are a potential reservoir of zoonotic Aeromonas spp. Moreover, what is worth highlighting, not only phenotypic but also determination of virulence-associated genes (by PCR) as well as phylogenetic analysis of the cphA gene have been performed. Some nucleotide sequences (gyrB and cphA) have been deposited in the GenBank database.

The writing is good and the data are presented properly in a clear and concise form.

The manuscript can be accepted for publication in Microorganism Journal after the following issues have been addressed. The manuscript needs Minor revision.

Response: We greatly appreciate the positive response and helpful advice from reviewer #1. Based on these comments, we have revised our manuscript.

Comments:

1) Materials and Methods, lines 99 and 110-111; and Results and Discussion; lines 183-184,

The isolated bacteria were identified by 16S rRNA sequencing ….

The authors used DNeasy Blood and Tissue kit to isolate bacterial genomic DNA and strains were identified by 16S rDNA/not rRNA sequencing.

Response: Thank you for the helpful comments. We have revised the manuscript accordingly.

2) Materials and Methods, lines 97-104; Please rearrange/reorder this fragment of the text. First isolated colonies should be selected, purified from single colonies subcloned, and then identified by 16S rDNA sequencing.

Response: Thank you for pointing this out. We have revised the manuscript accordingly.

3) Results and Discussion, Supplementary Table S1; The sizes of amplified PCR products(gene fragments) should be inserted in the Table S1.

Response: Thank you for this helpful comment. Based on this advice, we revised this table in the manuscript accordingly.

4) Results and Discussion, lines 206-221; The electrophoregrams showing the presence of amplified fragments of virulence-related genes should be inserted in the Supplementary.

Response: Thank you for this suggestion. Accordingly, we have added the electrophoregrams as Supplementary Table S3, showing the presence of amplified fragments of virulence-related genes as supplements.

5) Results and Discussion, lines 175-178; and lines 300-301; Therefore, the present study aimed to evaluate the incidence of zoonotic Aeromonas spp. In the Reviewer’s opinion this statement is exaggerated and should be modified. The incidence of potentially zoonotic Aeromonas sp.

Response: Thank you for pointing this out. Based on the advice of the reviewer, we revised these parts of the manuscript accordingly.

6) Results and Discussion, and lines 300-301; similarly as above.

Response: Thank you for these critical comments. Again, we have revised this sentence in the manuscript accordingly.

7) Table 4. R, I – under the Table should be explained.

Response: We apologize for this oversight. We have added these definitions to the table accordingly.

8) Results and Discussion, lines 267-268; Based on the absence of plasmid-mediated genetic determinants….Does this statement find support in the presented results? If does not please add appropriate literature positions.

Response: Thank you for pointing this out. Based on the advice of the reviewer, we revised the manuscript. Our corrections are shown in Lines 263–274. We have also added a new reference to improve the discussion and address this comment. The newly added reference is as follows:

66. de Been, M.; Lanza, V.F.; de Toro, M.; Scharringa, J.; Dohmen, W.; Du, Y.; Hu, J.; Lei, Y.; Li, N.; Tooming-Klunderud, A.; Heederik, D.J.J.; Fluit, A.C.; Bonten, M.J.M.; Willems, R.J.L.; de la Cruz, F.; van Schaik, W. Dissemination of cephalosporin resistance genes between Escherichia coli strains from farm animals and humans by specific plasmid lineages. PLoS Genetics 2014, 10, e1004776.

9) Results and Discussion. This paragraph should be modified. In the Reviewer’s opinion the results should be separated from the part constituting the discussion part.

Response: Thank you for this helpful comment. We agree with the reviewer to some extent, but if possible, we intend to maintain the current format of the manuscript. The purpose of this study was to investigate the two main risk factors (potential virulence and antibiotic resistance) of Aeromonas strains isolated from wild nutria in Korea, and we still think that combining the results and discussion sections of the manuscript will aid in helping potential readers investigating Aeromonas and/or wild animals to assess and interpret our results. Once again, we do appreciate your thoughtful advice.  

Editorial mistakes:

Title; and line 154, : Myocastor coypus, Aeromonas; should be written in italic font. 

Introduction; Please cut the instruction for introduction part of the manuscript.

Response: Thank you for pointing out these mistakes. We have corrected this in the revised manuscript.

Reviewer 2 Report

Review Wild Nutria (Myocastor coypus) is a Potential Reservoir of Carbapenem-resistant and Zoonotic Aeromonas spp. in Korea

The authors present a new manuscript about isolation and characterization of Aeromonas species from wild Nutria. The paper is well written and the data is presented in a standard and logical fashion. The manuscript is publishable with minor revisions.

The Discussion needs to be improved with a comparison of results between “Oliveira, M., Sales-Luís, T., Semedo-Lemsaddek, T., Ribeiro, T., Pedroso, N.M., Tavares, L. and Vilela, C.L., 2010. Antimicrobial Resistant Aeromonas Isolated from Eurasian Otters (Lutra lutra Linnaeus, 1758) in Portugal. Perspectives in Animal Ecology and Reproduction.” and their own data. In this manuscript the antibiotic resistance of isolated Aeromonas from Otters in Europe is described and although the authors of the current manuscript cite this paper, a comparative discussion is necessary.

Next, the authors cite “Lyon, W.J. and Milliet, J.B., 2000. Microbial flora associated with Louisiana processed frozen and fresh nutria (Myocastor coypus) carcasses. Journal of food science, 65(6), pp.1041-1045.” but also here a comparative discussion of their results is warranted. For example, does the media used for isolation matter, is Aeromonas really the most pathogenic species of the Nutria microbiome, does the isolation location (rectal, oral, wound) matter for isolation, etc.

Finally, the authors should discuss and reference

Navarro, A. and MartínezMurcia, A., 2018. Phylogenetic analyses of the genus Aeromonas based on housekeeping gene sequencing and its influence on systematics. Journal of applied microbiology, 125(3), pp.622-631

Lines 42-49 should be deleted.

Line 50, Gram should be capitalized. It is a person.

Author Response

<Response to reviewers: Ms. Ref. No. microorganisms-552864>

On behalf of all authors, I greatly appreciate the helpful advice and comments from ‘Microorganisms’ regarding our manuscript (microorganisms-552864) entitled "Wild Nutria (Myocastor coypus) is a Potential Reservoir of Carbapenem-resistant and Zoonotic Aeromonas spp. in Korea". According to the advice and comments of the reviewers, we have revised the manuscript, and all of our corrections are highlighted with yellow in the manuscript. We hope that the manuscript is now suitable for publication.

Ji Hyung Kim

-------------------------------------------------------------------------------------------------------------

REVIEWERS' COMMENTS TO AUTHORS

<Reviewer #2>

Review Wild Nutria (Myocastor coypus) is a Potential Reservoir of Carbapenem-resistant and Zoonotic Aeromonas spp. in Korea

The authors present a new manuscript about isolation and characterization of Aeromonas species from wild Nutria. The paper is well written and the data is presented in a standard and logical fashion. The manuscript is publishable with minor revisions.

Response: We do greatly appreciate the really helpful advices from the reviewer #2. Based on the comments of the reviewer #2, we faithfully revised our manuscript.

The Discussion needs to be improved with a comparison of results between “Oliveira, M., Sales-Luís, T., Semedo-Lemsaddek, T., Ribeiro, T., Pedroso, N.M., Tavares, L. and Vilela, C.L., 2010. Antimicrobial Resistant Aeromonas Isolated from Eurasian Otters (Lutra lutra Linnaeus, 1758) in Portugal. Perspectives in Animal Ecology and Reproduction.” and their own data. In this manuscript the antibiotic resistance of isolated Aeromonas from Otters in Europe is described and although the authors of the current manuscript cite this paper, a comparative discussion is necessary.

Response: Thank you for this helpful advice. Accordingly, we have added this reference (Oliveira et al., 2010) and revised the manuscript to discuss and compare this study. Moreover, we have also added new references to improve the discussion. Our corrections are shown in Lines 241–247. The newly added references are as follows:

63. Zottola, T.; Montagnaro, S.; Magnapera, C.; Sasso, S.; De Martino, L.; Bragagnolo, A.; D’Amici, L.; Condoleo, R.; Pisanelli, G.; Iovane, G.; Pagnini, U. Prevalence and antimicrobial susceptibility of Salmonella in European wild boar (Sus scrofa); Latium Region–Italy. Comp Immunol Microbiol Infect Dis 2013, 36, 161–168.

64. Jijón, S.; Wetzel, A.; LeJeune, J. Salmonella enterica isolated from wildlife at two Ohio rehabilitation centers. J Zoo Wildl Med 2007, 38, 409–414.

Next, the authors cite “Lyon, W.J. and Milliet, J.B., 2000. Microbial flora associated with Louisiana processed frozen and fresh nutria (Myocastor coypus) carcasses. Journal of food science, 65(6), pp.1041-1045.” but also here a comparative discussion of their results is warranted. For example, does the media used for isolation matter, is Aeromonas really the most pathogenic species of the Nutria microbiome, does the isolation location (rectal, oral, wound) matter for isolation, etc.

Response: Thank you again for pointing this out. Based on this comment, we have revised the manuscript accordingly. Our corrections are shown in Lines 188–191.

Finally, the authors should discuss and reference

Navarro, A. and MartínezMurcia, A., 2018. Phylogenetic analyses of the genus Aeromonas based on housekeeping gene sequencing and its influence on systematics. Journal of applied microbiology, 125(3), pp.622-631

Response: Thank you for this comment. Again, we have added this reference (Navarro and MartínezMurcia, 2018) and revised the manuscript accordingly. Our corrections are shown in Lines 177–181 and reference No. 49.

Lines 42-49 should be deleted.

Line 50, Gram should be capitalized. It is a person.

Response: Thank you for the helpful comments. Based on the advice of the reviewer, we have made these changes.